# Effects of Heat Treatment under Different Pressures on the Properties of Bamboo

**DOI:** 10.3390/polym15143074

**Published:** 2023-07-18

**Authors:** Dan Li, Shoulu Yang, Zhu Liu, Zhongwei Wang, Ning Ji, Jialei Liu

**Affiliations:** 1Guizhou Academy of Forestry, Guiyang 550005, China; lanwan83426@sina.com (D.L.); liuzhu9206@126.com (Z.L.);; 2School of Materials Science and Engineering, Central South University of Forestry and Technology, Changsha 410004, China; 3Institute of Environment, Sustainable Development in Agriculture, Chinese Academy of Agricultural Sciences, Beijing 100081, China

**Keywords:** bamboo, heat treatment, physical and chemical properties, pressure

## Abstract

To optimize the bamboo heat treatment process, the corresponding evolution rules under various heat treatment conditions must be determined. When the heat treatment time and temperature remained constant, the effects of different heat treatment pressures on the equilibrium moisture content, dimensional stability, mechanical properties, and chemical composition of bamboo were systematically investigated. In this experiment, bamboo without heat treatment was used as the control group. The experimental findings demonstrate the following: (1) The equilibrium moisture content of heat-treated bamboo gradually decreases with increasing treatment pressure. When the heat treatment pressure was set at 0.1, 0.15, 0.2, and 0.25 MPa, the bamboo’s equilibrium water content decreased to 12.1%, 11.7%, 9.9%, and 8.6%, respectively, while that of the control group was 13.8%. (2) The dimensional stability of bamboo was enhanced with increasing heat treatment pressure. At pressures of 0.1, 0.15, 0.2, and 0.25 MPa, the radial air-dry shrinkage rates of the heat-treated bamboo decreased to 3.4%, 3.4%, 2.6%, and 2.3%, respectively, while the tangential air-dry shrinkage rates reduced to 5.6%, 5.1%, 3.3%, and 3.0%. In comparison, the radial and tangential air-dry shrinkage rates of the control group were measured as 3.6% and 5.8%, respectively. Similarly, the radial and tangential full-dry shrinkage of bamboo exhibited a similar trend. (3) With the increase in heat treatment pressure, the bending strength and longitudinal compressive strength of bamboo exhibited an initial rise followed by a decline. When it was at heat treatment pressures of 0.1, 0.15, 0.2, and 0.25 MPa, the corresponding bending strengths of the heat-treated bamboo were measured as 41.2, 26.7, 22.4, and 20.4 MPa, respectively; while the longitudinal compressive strengths were recorded as 42.6, 38.1, 29.1, and 25.3 MPa. In comparison, the bending and longitudinal compressive strengths of the control group were measured as 39.8 and 38.5 MPa, respectively. It is evident that the optimal heat treatment pressure for bamboo is 0.1 MPa, resulting in a significant increase of 3.5% and 10.6% in bending strength and longitudinal compressive strength, respectively, compared to the control group. (4) Based on the FTIR and XRD patterns of bamboo samples, a range of physical and chemical transformations were observed during the heat treatment process, including cellulose adsorb water evaporation, hemicellulose and cellulose degradation, as well as acetyl group hydrolysis on the molecular chain of hemicellulose. These changes collectively impacted the physical and mechanical properties of bamboo.

## 1. Introduction

As a biomass material, bamboo is a raw structural material with excellent characteristics, including high strength, good toughness, and wear resistance [1]. However, bamboo has certain shortcomings. In particular, when the moisture content is lower than the fiber saturation point, the change in water absorption by the bamboo cell walls can cause dry shrinkage or wet expansion. The anisotropy of bamboo causes the extent of dry shrinkage and wet expansion to vary in dimensions in different directions, resulting in warping, deformation, and cracking. These drawbacks are regarded as the main obstacles to the use of bamboo as a building and industrial material [2]. In addition, the relatively high contents of carbohydrates and proteins in the tissues of bamboo are prone to insect infestation and mildew, and the corresponding tissue structure changes lead to the degradation of physical and mechanical properties [3]. Over the past several decades, many efforts have been made to modify bamboo to improve its utilization performance, and heat treatment is considered a green and environmentally friendly physical modification technique. Heat treatment methods have been extensively applied to wood modification, and studies on the heat treatment of wood have made some outstanding achievements. First, the color of wood materials can be changed by heat treatment to meet the practical demands for materials with different colors and to increase the value of the products. Secondly, numerous hydroxyl groups between or within the polymers of the wood removed water molecules and formed ether bonds in the condensation reaction during the heat treatment; thus, the moisture absorption of wood decreased, and the dimensional stability of wood can be improved [4,5]. Third, with the loss of water and the degradation of nutrients, such as carbohydrates, fats, and proteins, after heat treatment, the performance of wood against mildew, decay, and insects can be remarkably enhanced [6,7]. As a biomass material, bamboo has a porous cell wall structure similar to wood [8]. With changes in environmental humidity, the wet expansion or dry shrinkage of bamboo is a result of moisture absorption or desorption, respectively, leading to dimensional changes in the bamboo. Additionally, similar to wood, bamboo, with relatively high contents of various substances, such as carbohydrates, fats, and proteins, can be affected by mildew, decay, and insect infestation. Accordingly, the heat treatment method for wood can be utilized to modify bamboo. However, because bamboo is an anisotropic biomass material with an internode structure, when the heat treatment process is inappropriate, the cracks induced by uneven anisotropic shrinkage and the degradation of polymers cause the reducing of mechanical properties, which is detrimental to the usability of bamboo. Therefore, research on the changing properties of bamboo after heat treatment is significant for the development of bamboo applications.

Currently, the research on bamboo steam heat treatment primarily centers around investigating the impact of varying process conditions such as heat treatment duration, temperature, or other different processes on bamboo properties [9]. For example, the effect of temperature and time on the micromechanical properties of the bamboo cell wall, and the change in creep ratio after saturated steam heat treatment [10], etc. The majority of bamboo undergoes heat treatment at a higher temperature, such as 160 °C, 180 °C, or higher; however, there is a paucity of literature on the impact of steam pressure on bamboo performance at lower temperatures. This study proposes the effects of four pressure gradients on the physical and chemical properties of bamboo under medium and low temperature conditions, providing a theoretical foundation for regulating bamboo properties and optimizing heat treatment process parameters. Furthermore, in terms of economic costs, conducting further research on this topic would be valuable.

## 2. Materials and Methods

### 2.1. Experimental Material

#### 2.1.1. Bamboo

The material used in this study was the 5-year-old moso bamboo (Phyllostachys pubescens Mazel ex H. de Lebaie) harvested in Chishui City of Guizhou Province from China. The bamboo culms with the average dimensions of 1000 mm (length) × 13 mm (wall thickness) × 150 mm (diameter) were taken at a height of more than 800 mm from the roots. The bamboo culms were first split longitudinally into strips, and then its inner and outer skin were stripped off, resulting in a bamboo strip with dimension of 300 mm (length) × 20 mm (width) × 5 mm (thickness) along the longitudinal, radial, and tangential axes through a planing process as shown in Figure 1. 

#### 2.1.2. Test Instruments and Equipment

The following equipment was used: electric blast drying oven (101-3AB; Tianjin Test Instrument Co., Ltd., Tianjian, China), electronic balance (FA2004; Shanghai Jingke Balance, Shanghai, China), plant crusher (FW 400 A; Beijing Zhongxing Weiye Instrument Co., Ltd., Beijing, China), electronic universal mechanical testing machine (AG-IC100 kN; Shimadzu Co., Kyoto, Japan), Fourier-transform infrared spectrometer (Nicolet 670, Thermo Fisher, Waltham, MA, USA), X-ray diffractometer (D8 Advance, Bruker AXS, Karlsruhe, Germany), carbonization furnace, and steam generator.

### 2.2. Test Methods

#### 2.2.1. Heat Treatment of Bamboo

A bamboo strip measuring 300 mm × 20 mm × 5 mm was placed in a carbonization furnace, where it underwent heat treatment for one hour at a controlled temperature of 130 °C, while water steam was introduced into the furnace using a steam generator. By controlling the amount of water steam entering the carbonization furnace, the heat treatment pressure in the carbonization furnace was adjusted to 0.1, 0.15, 0.2, and 0.25 MPa. Moreover, the heat treatment pressure was maintained at a constant level throughout each heat treatment process. Bamboo without heat treatment was used as the control group (C). 

#### 2.2.2. Measurement of Equilibrium Moisture Content

The equilibrium moisture contents of the bamboo samples were using the “Test Method for Physical and Mechanical Properties of Bamboo” in the national Chinese standard, GB/T 15780-1995. The control group and the samples after heat treatment at various pressures (0.1, 0.15, 0.2, and 0.25 MPa) were placed into distilled water until they reached a saturated state. Subsequently, the samples were put into a box with a constant temperature of 20 °C and a relative humidity of 65% until their quality remained stable. The samples were then weighed (*m*_1_). The samples were completely dried in a drying box at 103 ± 2 °C and then weighed again (*m*_0_). The equilibrium moisture content (*E*) was calculated using Equation (1). A total of 35 repetitions were conducted for each pressure treated sample and control group.
(1)$$E=\frac{m_{1}-m_{0}}{m_{0}}\times 100$$

#### 2.2.3. Measurement of Dry Shrinkage

The tangential and radial shrinkage rates of the bamboo samples were tested using the “Test Method for Physical and Mechanical Properties of Bamboo” in the national Chinese standard, GB/T 15780-1995. The bamboo samples were processed by heat treatment under various pressures and the control group were cut to dimensions of 20 mm × 20 mm × 5 mm. After soaking in distilled water to a saturated state, the radial and tangential lengths were measured and denoted as L_0_. The samples were then put into a box at a constant temperature of 20 °C and a relative humidity of 65% until their quality remained stable; after which, the radial and tangential lengths were measured and denoted as L_1_. After drying in an oven at 103 ± 2 °C to a constant weight, the radial and tangential lengths were measured again and denoted as L_2_. The air-drying shrinkage rate was obtained using Equation (2), and the full-drying shrinkage rate was determined using Equation (3). A total of 35 repetitions were conducted for each pressure treated sample and control group.
(2)$$\mathrm{Radial}/\tan\mathrm{gential}\;\mathrm{air}\text{-}\mathrm{dry}\;\mathrm{shrinkage}\;\mathrm{rate}=\frac{L_{0}-L_{1}}{L_{0}}\times 100$$
(3)$$\mathrm{Radial}/\tan\mathrm{gential}\;\mathrm{full}\text{-}\mathrm{dry}\;\mathrm{shrinkage}\;\mathrm{rate}=\frac{L_{0}-L_{2}}{L_{0}}\times 100$$

#### 2.2.4. Measurement of Mechanical Properties 

The bamboo samples processed by heat treatment under various pressures and the control group were cut into dimensions of 200 mm × 20 mm × 5 mm. The bending strength and longitudinal compressive strength of the bamboo samples were determined using the “test method for physical and mechanical properties of bamboo” in the national Chinese standard GB/T 15780-1995. The bending and longitudinal compressive strengths of the samples were calculated using Equations (4) and (5), respectively. A total of 35 repetitions were conducted for each pressure treated sample and control group.
(4)$${б}_{\mathrm{bw}}=\frac{3\times P_{max}\times L}{2\times b\times h^{2}}$$
(5)$${б}_{w}=\frac{P_{max}}{b\times t}$$
where *б*_bw_ is the bending strength of a sample with a moisture content of W%, in MPa; *б*_w_ is the longitudinal compressive strength of a sample with a moisture content of W%, in MPa; *P_max_* is the failure load; *L* is the distance between two holders (120 mm); *b* is the width of the sample, in mm; *t* is the thickness of the sample, in mm; and h is the height of the sample, in mm.

#### 2.2.5. Evaluation of Chemical Properties

The bamboo samples processed by heat treatment and the control group were crushed into a 200-mesh powder with a plant crusher. After the samples were completely dried, their microstructural characteristics were analyzed using Fourier-transform infrared spectrometry (FTIR) and X-ray diffractometry (XRD). The scanning wavenumber of the FTIR analysis was within the range of 4000 cm^−1^~400 cm^−1^, the scanning angle of the XRD analysis was within the range of 5–50°, and the scanning speed was set to 8°/min.

## 3. Results and Analysis

### 3.1. Influence of Different Pressure Heat Treatment on the Equilibrium Moisture Content of Bamboo

Figure 2 shows the equilibrium moisture content of the bamboo samples after heat treatment at different pressures. It was found that the equilibrium moisture content of the heat-treated bamboo samples gradually decreased with the increase in heat treatment pressure within the carbonization furnace. There was no significant change in the equilibrium moisture content between 0.1 and 0.15 MPa. In contrast, the equilibrium moisture content was significantly different between 0.2 and 0.25 MPa. Compared to the control group (the equilibrium moisture content of the control group was 13.8%), as the pressure increased, the equilibrium moisture content of the bamboo processed by heat treatment decreased to 12.1%, 11.7%, 9.9%, and 8.6%, respectively. Under low-temperature hydrothermal conditions, hemicellulose, which has the strongest water absorption among the three major components of the cell wall, was degraded, and the glucoside bond on its molecular chain was destroyed, resulting in the decrease in hemicellulose content. Meanwhile, the acetyl groups in the polysaccharide molecular chains of hemicellulose underwent hydrolysis, and the acetic acid produced by this reaction promoted the degradation of hemicellulose. Due to the occurrence of the above reaction, the number of hydrophilic groups such as hydroxyl and carbonyl groups decreases, leading to less hygroscopicity of the bamboo. In addition, as the heat treatment pressure increases, hemicellulose degradation and acetyl hydrolysis gradually intensify, leading to a gradual decrease in the amount of hydroxyl and carbonyl groups in bamboo. Therefore, with the increase in heat treatment pressure, the equilibrium moisture contents of bamboo gradually decreased compared with the control group.

### 3.2. Influence of Different Pressure Heat Treatments on the Dimensional Stability of Bamboo 

Table 1 lists the tangential and radial air-dry shrinkage rates of bamboo. The tangential and radial air-dry shrinkage rates of the bamboo showed a gradually decreasing trend after heat treatment, and the dry shrinkage rate showed little change between 0.1 and 0.15 MPa. Compared to the control group (the tangential and radial rates of air-dry shrinkage for the control group were 5.8% and 3.6%, respectively), the tangential and radial air-dry shrinkage rates of bamboo subjected to heat treatment at 0.1 MPa were reduced to 5.6% and 3.4%, respectively. With the increase in pressure, the tangential shrinkage rate of bamboo after heat treatment clearly changed between 0.15 and 0.25 MPa. When the pressure was set to 0.25 MPa, the tangential and radial air-dry shrinkage rates of heat-treated bamboo decreased to 3.0% and 2.3%, respectively. Similarly, the heat-treated bamboo showed consistent trends in both tangential and radial full-dry shrinkage rates (Table 2). It can be seen that the dimensional stability of the bamboo was dramatically improved after heat treatment. The mechanisms responsible for this improvement are mainly dominated by the following aspects. On the one hand, the heat treatment resulted in the degradation of bamboo hemicellulose, leading to a reduction in the amount of hydroxyl groups present. Meanwhile, hydrolysis of the acetyl groups in hemicellulose polysaccharide molecular chains resulted in a reduction in carbonyl group content. The decrease in hydrophilic groups, such as hydroxyl and carbonyl, led to a reduction in the hygroscopicity and dry shrinkage of bamboo. Consequently, the dimensional stability of bamboo was enhanced [11]. On the other hand, during heat treatment, acetic acid produced by the hydrolysis of acetyl groups in bamboo hemicellulose entered the amorphous region of cellulose and promoted degradation of cellulose molecules in this region, thereby increasing the relative proportion of crystalline regions within cellulose and improving its relative crystallinity. This change is conducive to enhancing the dimensional stability of bamboo [12]. Simultaneously, during heat treatment, the evaporation of adsorbed water in bamboo reduced the distance between cellulose molecular chains in the amorphous region, thereby increasing intermolecular forces and enhancing orientation of cellulose molecules in heat-treated bamboo’s amorphous region, ultimately improving dimensional stability [13].

### 3.3. Influence of Different Pressure Heat Treatments on the Mechanical Properties of Bamboo

As shown in Figure 3, after the heat treatment of bamboo, the bending strength and longitudinal compressive strength first increased and then decreased, reaching maximum values when the pressure was set to 0.1 MPa. The variations in the mechanical properties of bamboo are mainly derived from changes in its physical and chemical properties during heat treatment under different pressures. When the pressure was set to 0.1 MPa, the bending strength and longitudinal compressive strength of the bamboo increased by 3.5% and 10.6%, respectively, compared to the control group. Based on the FTIR spectra analysis of bamboo samples subjected to heat treatment, it was observed that the degradation level of hemicellulose remained limited under this pressure condition (0.1 MPa), which had a negligible impact on the mechanical properties of bamboo specimens. The primary change observed under this heat treatment condition was the evaporation of adsorbed water in bamboo. The evaporation of adsorbed water in bamboo shortens the distance between cellulose molecular chains in the amorphous region, thereby increasing intermolecular forces and promoting tighter arrangement and orderliness. Consequently, this reduces the equilibrium moisture content of bamboo to a certain extent. Below the fiber saturation point, the mechanical properties of bamboo exhibit a negative correlation with its moisture content [14]. With the further increase in pressure, the bending and longitudinal compressive strength after heat treatment showed a decreasing trend. Additionally, the extent of hemicellulose degradation in bamboo is exacerbated by elevated heat treatment pressures, leading to the disruption of linkages between hemicellulose, lignin, and cellulose and a reduction in joint nodes, ultimately resulting in intercellular layer cleavage [15]. However, the temperature remains relatively stable during heat treatment, and the increase in mechanical strength resulting from adsorbed water evaporation is limited. Consequently, bamboo’s mechanical properties eventually deteriorate under the combined effects of heat and pressure. In this series of heat treatment experiments, the bending strength and longitudinal compressive strength of bamboo were affected by the combined effects of decreasing water content and hemicellulose degradation.

### 3.4. Influence of Different Pressure Heat Treatments on the Chemical Composition of Bamboo

#### 3.4.1. Influence of Different Pressure Heat Treatments on the FTIR Characteristics of Bamboo 

Figure 4 shows the FTIR spectra of bamboo samples processed by heat treatment under different pressures and bamboo samples from the control group at the wave number of 4000~400 cm^−1^. The main absorption peaks of the FTIR spectra were assigned as follows: the stretching vibration of the hydroxyl group (–OH) at 3420 cm^−1^; the bending vibration of the hydroxyl group (O–H) of adsorbed water in cellulose at 1630 cm^−1^; the stretching vibration of the carbonyl group (C=O) of hemicellulose ester at 1736 cm^−1^; and the stretching vibrations of β-anhydroglucose bonds (C–C–O and C–O–C) at 1164 cm^−1^ and 1035 cm^−1^, respectively. As depicted in Figure 4, the peak intensity of the C–H absorption at 1382 cm^−1^ remained relatively stable. The bending vibration of the C–H absorption peak was utilized as a reference to determine the relative intensities of other absorption peaks at 3420 cm^−1^, 1736 cm^−1^, 1630 cm^−1^, 1164 cm^−1^, and 1035 cm^−1^, respectively. After heat treatment, the peak intensities of the hydroxyl group in bamboo (3420 cm^−1^) and the hydroxyl group of adsorbed water in cellulose (1630 cm^−1^) significantly decreased compared to those of the control group as shown in Figure 4 and Table 3. This suggests that the evaporation of adsorbed water in bamboo occurred during heat treatment. When the pressure increased from 0.1 to 0.25 MPa, there was no significant change in the intensities of absorption peaks at 3420 cm^−1^ and 1630 cm^−1^, indicating that the variation in pressure had a limited impact on such change processes. It can be deduced that the heat treatment temperature was still the main factor affecting this change. The peak intensity of the carbonyl group (1736 cm^−1^) in the hemicellulose ester of the heat-treated bamboo sample showed a decreasing trend compared to that in the control group. This suggests that hemicellulose, which exhibits inferior thermal stability, has undergone degradation under hydrothermal conditions resulting in a decrease in its content during heat treatment. Concurrently, the acetyl groups present within the polysaccharide chains of hemicellulose undergo hydrolysis and generate acetic acid [16], which subsequently penetrates into the amorphous region of cellulose and promotes degradation of cellulose molecules within this domain. The decrease in the intensity of characteristic peaks at 1164 cm^−1^ and 1035 cm^−1^, corresponding to C–C–O and C–O–C bonds in β-glucose anhydride, respectively, indicated the occurrence of this process. With an increase in heat treatment pressure, the concentration of carbonyl groups in hemicellulose esters decreased, indicating that increased pressure promoted hemicellulose degradation. Notably, at pressures of 0.2 and 0.25 MPa, hemicellulose degradation was significantly exacerbated during heat treatment. Among the above changes, the evaporation of adsorbed water in bamboo shortens the distance between the cellulose molecular chains in the amorphous region of cellulose and forms new hydrogen bonds so that the cellulose molecular chains are more closely arranged and ordered [17], which is conducive to an improvement in the bending strength and longitudinal compressive strength of bamboo. Additionally, the degradation of hemicellulose and acetyl group hydrolysis reactions result in a reduction in hydrophilic groups, such as hydroxyl and carbonyl groups, thereby weakening the moisture absorption capacity of bamboo while enhancing its dimensional stability. However, hemicellulose degradation is not conducive to the mechanical properties of bamboo. Among the three main components of the bamboo cell wall, hemicellulose acts as a binding agent, wich can fill the gaps between fibers and microfibers [18]. The reduction in hemicellulose weakens the cell wall network skeleton formed by cellulose, thereby reducing the mechanical properties of bamboo. Table 3 shows that, when the pressure was set at 0.1 MPa, only a small amount of hemicellulose was degraded during heat treatment, and the main changes could be ascribed to the evaporation of adsorbed water in the bamboo. Compared with the bamboo in the control group, bamboo in the 0.1 MPa group showed increased bending strength and longitudinal compressive strength. When the pressures were set at 0.15, 0.2, and 0.25 MPa, the degradation of hemicellulose was gradually aggravated, which caused embrittlement of the bamboo and a decrease in its mechanical properties.

#### 3.4.2. Influence of Different Pressure Heat Treatments on the XRD Characteristics of Bamboo

The relative crystallinity of bamboo cellulose can be expressed as a percentage of the crystalline zone relative to the total cellulose content of the bamboo. The mechanical properties of bamboo, such as modulus of elasticity and tensile strength, are closely related to its crystallinity, which also affects the hygroscopicity and dimensional stability of bamboo to some extent. Figure 5 shows the influence of different heat treatment pressures on the XRD characteristics of bamboo. The Segal method was adopted to calculate the relative crystallinity of cellulose in the bamboo samples processed by heat treatment under different pressures, and the results are listed in Table 4. Compared to the control group, the (002) crystal plane diffraction peaks of the bamboo samples processed by heat treatment under different pressures were concentrated within the range of 21.8–22.0°. This indicates that the heat treatment of the bamboo had no evident impact on the crystalline zone of cellulose; thus, there was no significant change in the distance between the crystal layers in the crystalline zone. Nevertheless, it can also be seen that the heat treatment of bamboo had a certain effect on the relative crystallinity of cellulose. After heat treatment, as the pressure increased, the relative crystallinity of the cellulose reached 46.5, 46.8, 47.1, and 48.2% (as Table 4 shows). Compared to the control group, the bamboo samples processed by heat treatment showed increased relative crystallinity of cellulose to different degrees. During the process of heat treatment, hydrolysis of the acetyl groups within the polysaccharide molecular chains of hemicellulose occured, resulting in the production of acetic acid. This acid then entered into the amorphous region of cellulose and promoted degradation of cellulose specifically within this area. As a result, there was an increase in the relative proportion of crystalline zones within cellulose [19].

We believe that the increase in relative crystallinity of cellulose is a result of both the physical and chemical changes occurring during heat treatment. On the one hand, the evaporation of adsorbed water in the amorphous region of cellulose reduces the distance between cellulose molecular chains, leading to the formation of new hydrogen bonds. This results in an increase in intermolecular forces and a more tightly arranged and ordered structure, ultimately resulting in an increase in relative crystallinity of bamboo cellulose. On the other hand, the acetic acid produced by the hydrolysis of acetyl groups in hemicellulose polysaccharide chains infiltrates into the amorphous region of cellulose, thereby promoting degradation in this area and increasing the relative proportion of crystalline cellulose zones. As evidenced by the data presented in Table 4, the heat treatment applied under this experimental condition primarily targets the amorphous region of bamboo cellulose, while leaving the crystalline region unaffected. This finding is consistent with previous research on high temperature heat treatments of bamboo. From an economic standpoint, this study highlights the potential value of further exploring medium and low temperature heat treatment processes.

## 4. Conclusions

In this study, the heat treatment time and temperature were kept constant while systematically investigating the effects of different pressures on bamboo’s equilibrium moisture content, dimensional stability, mechanical properties, and chemical composition. The experimental results demonstrated the following: (1) According to the FTIR and XRD spectra of bamboo samples, physical and chemical changes occurred during heat treatment in this experiment. These changes included the release of adsorbed water in bamboo, degradation of hemicellulose and cellulose, as well as hydrolysis of acetyl groups in the molecular chains of hemicellulose. Based on the information obtained from FTIR spectrum analysis, it was observed that the amount of adsorbed water in bamboo samples decreased under each heat treatment pressure condition compared to the control group. However, there was no significant change in the degree of decrease in adsorbed water when increasing heat treatment pressure. Therefore, we concluded that the heat treatment temperature was the primary factor influencing the release of adsorbed water of bamboo in this experiment. However, the extent of hemicellulose and cellulose degradation as well as acetyl group hydrolysis within hemicellulose molecular chains were enhanced with increasing heat treatment pressure. (2) With the increase in heat treatment pressure, there is a gradual decline in the equilibrium moisture content of bamboo. The physical and chemical changes in this experiment weaken the hygroscopic properties of bamboo, with hemicellulose degradation and acetyl group hydrolysis being the main influencing factors. These two reactions resulted in a decrease in the amount of hydrophilic groups, such as hydroxyl and carbonyl groups, in bamboo samples after heat treatment, thereby reducing the hygroscopic properties of the samples. (3) With the increase in heat treatment pressure, the dimensional stability of bamboo was improved; this is also due to the occurrence of the above physical and chemical changes, which weaken the moisture absorption ability of bamboo and increase the relative crystallinity. It can be seen that, without taking the mechanical properties of bamboo as the main consideration, heat treatment of bamboo in a pressurized steam atmosphere is an effective means of enhancing its dimensional stability. (4) With the increase in heat treatment pressure, the bending strength and longitudinal compressive strength of bamboo initially increased before subsequently decreasing. This change was jointly affected by the release of adsorbed water in bamboo and the degradation of hemicellulose and cellulose. These physical and chemical changes reduce the equilibrium water content while increasing the relative crystallinity of bamboo, which is conducive to an increase in its mechanical strength. However, the degradation of hemicellulose and cellulose simultaneously resulted in the destruction of the bamboo cell wall’s skeletal structure, rendering it fragile and ultimately reducing its mechanical strength. When subjected to a heat treatment pressure of 0.1 MPa, only a minimal amount of hemicellulose and cellulose underwent degradation during the process; instead, release of adsorbed water within the bamboo is primarily responsible for any changes observed at this stage—which ultimately serves to enhance its mechanical strength. At this stage, the bamboo exhibits its highest bending strength and longitudinal compressive strength, measuring 41.2 MPa and 42.6 MPa, respectively. As heat treatment pressure increases further, hemicellulose and cellulose degradation intensifies, leading to greater damage to the bamboo cell wall and a gradual decrease in both bending strength and longitudinal compressive strength.

## Figures and Tables

**Figure 1 polymers-15-03074-f001:**
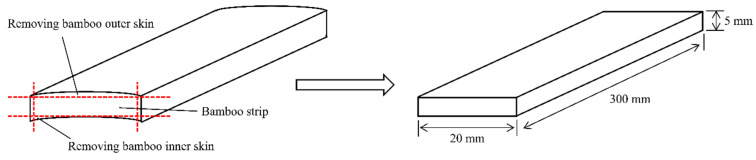
Diagram of bamboo sample preparation.

**Figure 2 polymers-15-03074-f002:**
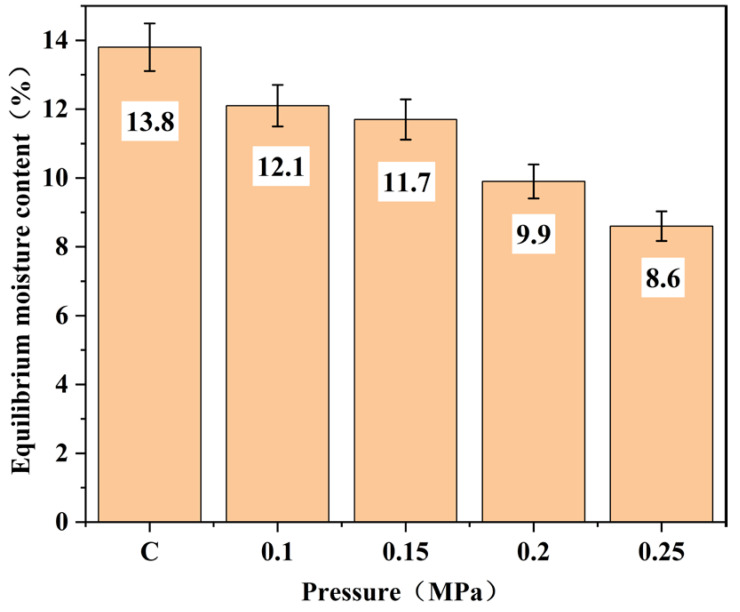
Equilibrium moisture contents of bamboo samples processed by heat treatment under different pressures.

**Figure 3 polymers-15-03074-f003:**
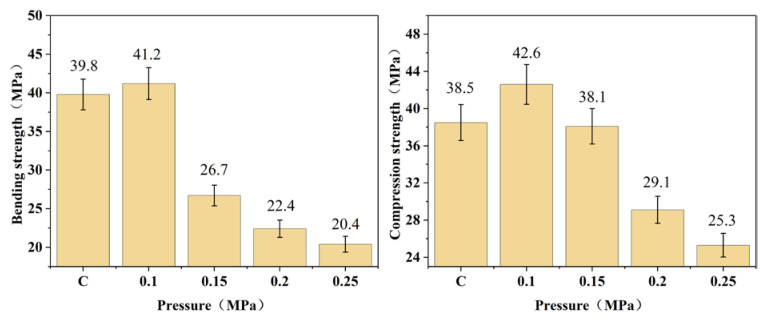
Mechanical properties of bamboo samples processed by heat treatment under different pressures.

**Figure 4 polymers-15-03074-f004:**
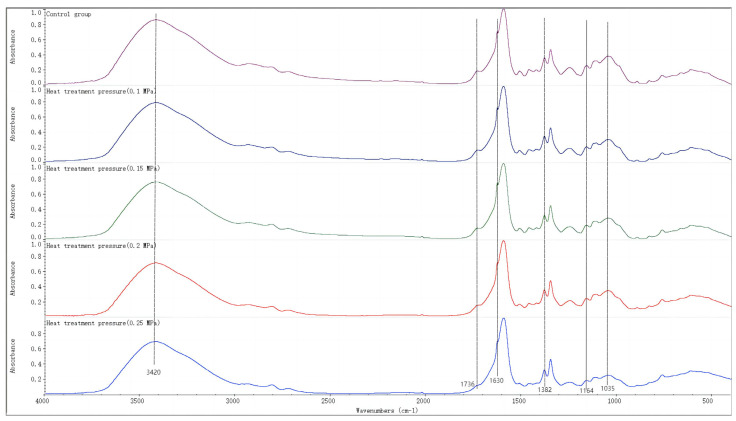
Fourier-transform infrared spectra of bamboo samples processed by heat treatment under different pressures and control bamboo samples.

**Figure 5 polymers-15-03074-f005:**
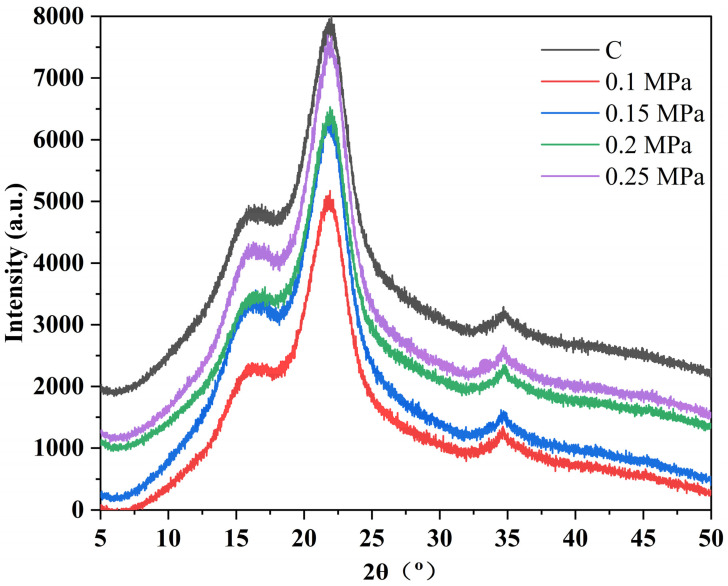
X-ray diffraction patterns of control bamboo samples and bamboo samples processed by heat treatment under different pressures.

**Table 1 polymers-15-03074-t001:** Radial/tangential air-dry shrinkage rates of bamboo samples processed by heat treatment under different pressures.

Pressure (MPa)	Radial Air-Dry Shrinkage Rate (%)	Tangential Air-Dry Shrinkage Rate (%)
Control group (C)	3.6 (0.03)	5.8 (0.05)
0.1	3.4 (0.03)	5.6 (0.04)
0.15	3.4 (0.04)	5.1 (0.03)
0.2	2.6 (0.04)	3.3 (0.04)
0.25	2.3 (0.06)	3.0 (0.07)

The value in the brackets represents the standard deviation.

**Table 2 polymers-15-03074-t002:** Radial/tangential full-dry shrinkage rates of bamboo samples processed by heat treatment under different pressures.

Pressure (MPa)	Radial Full-Dry Shrinkage Rate (%)	Tangential Full-Dry Shrinkage Rate (%)
Control group (C)	4.8 (0.11)	7.0 (0.14)
0.1	4.6 (0.19)	6.1 (0.10)
0.15	3.1 (0.31)	4.3 (0.23)
0.2	2.4 (0.16)	2.1 (0.08)
0.25	2.0 (0.18)	2.1 (0.20)

The value in the brackets represents the standard deviation.

**Table 3 polymers-15-03074-t003:** Relative intensities of characteristic peaks of bamboo samples processed by heat treatment under different pressures and control bamboo samples.

Wave Number /cm^−1^	Control Group (C)		Heat Treatment under Different Pressures							
			0.1 MPa		0.15 MPa		0.2 MPa		0.25 MPa	
	Intensity	Relative Amount	Intensity	Relative Amount	Intensity	Relative Amount	Intensity	Relative Amount	Intensity	Relative Amount
1382	0.112	1.00	0.124	1.00	0.113	1.00	0.116	1.00	0.111	1.00
3420	0.726	6.48	0.670	5.40	0.644	5.70	0.634	5.47	0.625	5.63
1736	0.030	0.27	0.027	0.22	0.022	0.19	0.015	0.13	0.003	0.03
1164	0.075	0.67	0.059	0.48	0.055	0.49	0.051	0.44	0.034	0.31
1035	0.169	1.51	0.133	1.07	0.123	1.09	0.118	1.02	0.073	0.66

**Table 4 polymers-15-03074-t004:** Crystallization characteristics of control bamboo samples and bamboo samples processed by heat treatment under different pressures.

Heat Treatment Under Different Pressures (MPa)	(002) Crystal Plane Angle (o)	Relative Crystallinity (%)
Control group (C)	22.0	45.1
0.1	21.8	46.5
0.15	21.9	46.8
0.2	21.9	47.1
0.25	22.0	48.2

## Data Availability

Supplemental data can be provided upon reasonable request.

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
