# Peer review of "Effects of Heat Treatment under Different Pressures on the Properties of Bamboo"

_polymers, 2023, doi:10.3390/polym15143074_

Round 1

Reviewer 1 Report

See comments in the attached file.

See above.

Stype of writing is complicated and should be revised for better straight forward mode and readability. Partly it is not clear what the authors wanted to express.

Author Response

The parts of the paper

Comments of Reviewer 1

Our actions

Abstract 1. Abstract: The authors wrote: “The experimental results demonstrated that the equilibrium moisture content of bamboo decreased with a decrease in the number of free hydroxyl groups, and it also exhibited a gradually decreasing trend with an increase in the heat treatment pressure.” The abstract should be rewritten, including more details about the experimental results including the values or percentage of decrease or increase of the bamboo’s properties.

To optimize the bamboo heat treatment process, the corresponding evolution rules under various heat treatment conditions must be determined. When the heat treatment time and temperature remained constant, the effects of various vapour pressures on the equilibrium moisture content, dimensional stability, mechanical properties, and chemical composition of bamboo were systematically investigated. The experimental findings demonstrate that: (1) The equilibrium moisture content of heat-treated bamboo gradually decreases with increasing treatment pressure. When the heat treatment pressure was set at 1.0, 1.5, 2.0 and 2.5 MPa, the bamboo's equilibrium water content decreased to 12.1%, 11.7%, 9.9% and 8.6%, respectively. (2) The dimensional stability of bamboo was enhanced with increasing heat treatment pressure. At pressures of 1.0, 1.5, 2.0 and 2.5 MPa, the radial air-dry shrinkage rates decreased to 3.41%, 3.35%, 2.57% and 2.34%, respectively, while the tangential air-dry shrinkage rates reduced to be at 5.63%, 5.11%, 3.27% and 2.96%. Similarly, the radial and tangential full-dry shrinkage of bamboo exhibited a similar trend. (3) With the increase of heat treatment pressure, the bending strength and longitudinal compressive strength of bamboo exhibited an initial rise followed by a decline. When it was at heat treatment pressures of 1.0, 1.5, 2.0 and 2.5 MPa, the corresponding bending strengths were measured as 41.2, 26.7, 22.4 and 20.4MPa respectively; while the longitudinal compressive strengths were recorded as 42.6, 38.1, 29.1 and 25.3MPa. It is evident that the optimal heat treatment pressure for bamboo is 1.0 MPa, resulting in a significant increase of 3.5% and 10.6% in bending strength and longitudinal compressive strength, respectively, compared to the control group. (4) Based on the FTIR and XRD patterns of bamboo samples, a range of physical and chemical transformations were observed during the heat treatment process, including cellulose adsorb water evaporation, hemicellulose and cellulose degradation, as well as acetyl group hydrolysis on the molecular chain of hemicellulose. These changes collectively impacted the physical and mechanical properties of bamboo.

Introduction 2. Introduction: In the last paragraph, the main goal of the research and the advancement of the frontier of knowledge intended must be shown clearly.

Currently, the research on bamboo steam heat treatment primarily centers around investigating the impact of varying process conditions such as heat treatment duration, temperature or other different process on bamboo properties [9]. For example, the effect of temperature and time on the micromechanical properties of bamboo cell wall, and the change of creep ratio after saturated steam heat treatment [10], etc . And the majority of them undergo heat treatment at a higher temperatures, such as 160℃, 180℃ or higher. However, there is a paucity of literature on the impact of steam pressure on bamboo performance at lower temperatures. This study proposes the effects of four pressure gradients on the physical and chemical properties of bamboo under medium and low temperature conditions, providing a theoretical foundation for regulating bamboo properties and optimizing heat treatment process parameters. Furthermore, in terms of economic costs, conducting further research on this study would be valuable.

Materials and methods 3. Materials and methods: Where was bamboo collected? This information is essential, and the authors should include it in the manuscript and, if possible, geographical coordinates.

The material used in this study was the 5-year-old moso bamboo (Phyllostachys pubescens Mazel ex H. de Lebaie) harvested in Chishui City of Guizhou Province from China. The bamboo culms with the average dimensions of 1000 mm (length)×13 mm (wall thickness)×150 mm (diameter) were taken at a height of more than 800 mm from the roots. The bamboo culms was firstly split longitudinally into some strips, and then stripped of its inner and outer skin, resulting a bamboo strip with dimension of 300 mm (length)×20 mm (width)×5 mm (thickness) along the longitudinal, radial and tangential axes through planing process as shown in Figure 1-1.

4. Materials and methods: How many repetitions were used in the experiments? Duplicate, triplicate, or other. This information must appear in the manuscript.

35 repetitions for each pressure treated sample and control group

5. Is the Equation (1) correct? What is the number or parameter after x?

Results and analysis 6. In Figure 2-3, the authors could include the main groups to illustrate better the FTIR spectra.  
  7. The conclusion is consistent with the evidence and arguments.

  8. The references are appropriate.

Reviewer 2 Report

The paper presents a study about the effects of heat treatment under different pressures on the properties of bamboo. So, it brings an essential contribution to the biomass material sector. However, before consideration for publication, specific points should be reviewed.

1) Abstract: The authors wrote: “The experimental results demonstrated that the equilibrium moisture content of bamboo decreased with a decrease in the number of free hydroxyl groups, and it also exhibited a gradually decreasing trend with an increase in the heat treatment pressure.” The abstract should be rewritten, including more details about the experimental results including the values or percentage of decrease or increase of the bamboo’s properties.

2) Introduction: In the last paragraph, the main goal of the research and the advancement of the frontier of knowledge intended must be shown clearly.

3) Materials and methods: Where was bamboo collected? This information is essential, and the authors should include it in the manuscript and, if possible, geographical coordinates.

4) Materials and methods: How many repetitions were used in the experiments? Duplicate, triplicate, or other. This information must appear in the manuscript.

5) Is the Equation (1) correct? What is the number or parameter after x?

6) In Figure 2-3, the authors could include the main groups to illustrate better the FTIR spectra.

7) The conclusion is consistent with the evidence and arguments.

 8) The references are appropriate. 

 Minor editing of English language required. 

Author Response

Dear reviewer, thank you so much for your valuable suggestions and insightful comments on our paper. We have carefully addressed all of your inquiries and made the necessary revisions to the manuscript based on your feedback.

Line in the article

Comments of Reviewer 2

Our actions

12

... various vapour pressures ...

When the heat treatment time and temperature remained constant, the effects of various vapour pressures on the equilibrium moisture content, dimensional stability, mechanical properties, and chemical composition of bamboo were systematically investigated.
26

..., when ...

In particular, when the moisture content is lower than the fiber saturation point, the change in water absorption by the bamboo cell walls can cause dry shrinkage or wet expansion.

39

A color cannot be improved, but only changed. Often darker color due to heat treatment is not prefered.

First, the color of wood materials can be changed by heat treatment to meet the practical demands for materials with different colors and to increase the value of the products.

42

How can water molecules split off from hydroxyl groups?

Secondly, numerous hydroxyl groups between or within the polymers of the wood removed water molecules and formed ether bonds in the condensation reaction during the heat treatment, thus, the moisture absorption of wood decreased and the dimensional stability of wood can be improved.

43

From which reaction do the ether bonds come from?

56

What does "utilization rate" mean here?

The statement about improving the “utilization rate” of bamboo is not very appropriate, so we have made the following modifications:

However, because bamboo is an anisotropic biomass material with an internode structure, when the heat treatment process is inappropriate, the cracks induced by uneven anisotropic shrinkage and the degradation of polymers cause the reducing of mechanical properties , that is detrimental to the usability of bamboo. Therefore, research on the changing properties of bamboo after heat treatment is significant for the development of bamboo applications.

63

This must be explained more in detail. Is there the risk that such a pretreatment destroys partly the original structure of the bamboo. What was the diameter of the bamboo stems felled, in comparison with the dimensions of the plate? What had been the conditionsfor this step?

The material used in this study was the 5-year-old moso bamboo (Phyllostachys pubescens Mazel ex H. de Lebaie) harvested in Chishui City of Guizhou Province from China. The bamboo culms with the average dimensions of 1000 (length)×13 (wall thickness)×150 mm (diameter) were taken at a height of more than 800 mm from the roots. The bamboo culms was firstly split longitudinally into some strips, and then stripped of its inner and outer skin, resulting a bamboo strip with dimension of 300 (length)×20 (width)×5 mm (thickness) along the longitudinal, radial and tangential axes through planing process as shown in Figure 1-1.

76

Was this a steam-saturated atmosphere?

A bamboo strip with dimensions 300 mm × 20 mm × 5 mm was placed in a carbonization furnace and subjected to heat treatment using steam-saturated atmosphere.

83

What is the reason for the step in distilled water before storage at the climate 20/65?

We wanted the bamboo samples processed by heat treatment and the control group to be in the same state before pretreatment.

101

were

The bamboo samples processed by heat treatment under various pressures and the control group were cut into dimensions of  200 mm × 20 mm × 5 mm.

127

in absolute numbers

Compared to the control group(The equilibrium moisture content of control group was 13.8%), as the pressure increased, the equilibrium moisture content of the bamboo processed by heat treatment decreased to 12.1%, 11.7%, 9.9%, and 8.6%, respectively.

129

... to less hygroscopicity ...

According to the reviewer's opinion, we re-examined the data of this experiment and found no obvious evidence to support the production of ether bonds in this experiment, so we re-analyzed this part according to the experimental data:

Under low-temperature hydrothermal conditions, hemicellulose, which has the strongest water absorption among the three major components of the cell wall, is degraded, and the content of hemicellulose decreases. In the meanwhile, the acetyl groups in the polysaccharide molecular chains of hemicellulose undergone hydrolysis. Due to the occurrence of the above reaction, the amount of hydrophilic groups such as hydroxyl and carbonyl groups decreases, leading to less hygroscopicity of the bamboo. In addition, with the increase of heat treatment pressure, hemicellulose degradation and acetyl hydrolysis gradually intensified, and the loss of hydroxyl and carbonyl groups gradually increased. Therefore, with the increase of heat treatment pressure, the equilibrium moisture contents of bamboo gradually decreased compared with the control group.

130

The

131

Why are ether bonds formed when cellulose is degraded?

152

during

Table 2-1 lists the tangential and radial air-dry shrinkage rates of bamboo. The tangential and radial air-dry shrinkage rates of the bamboo showed a gradually decreasing trend after heat treatment, and the dry shrinkage rate showed little change between 1.0 and 1.5 MPa. Compared to the control group(The tangential and radial rates of air-dry shrinkage for the control group were 5.80% and 3.55%, respectively.), the tangential and radial air-dry shrinkage rates of bamboo subjected to heat treatment at 1.0 MPa were reduced to 5.63% and 3.41%, respectively. With the increase in pressure, the tangential shrinkage rate of bamboo after heat treatment clearly changed between 1.5 and 2.5 MPa. When the pressure was set to 2.5 MPa, the tangential and radial air-dry shrinkage rates of heat-treated bamboo decreased to 2.96% and 2.34%, respectively. Similarly, the heat-treated bamboo showed consistent trends in both tangential and radial full-dry shrinkage rates. (Table 2-2). It can be seen that the dimensional stability of the bamboo was dramatically improved after heat treatment. The mechanisms responsible for this improvement are mainly dominated by the following aspects. On the one hand,The heat treatment resulted in the degradation of bamboo hemicellulose, leading to a reduction in the amount of hydroxyl groups present. Meanwhile, hydrolysis of the acetyl groups in hemicellulose polysaccharide molecular chains resulted in a reduction of carbonyl group content. The decrease in hydrophilic groups, such as hydroxyl and carbonyl, led to a reduction in the hygroscopicity and dry shrinkage of bamboo. Consequently, the dimensional stability of bamboo was enhanced. On the other hand, during heat treatment, acetic acid produced by the hydrolysis of acetyl groups in bamboo hemicellulose entered the amorphous region of cellulose and promoted degradation of cellulose molecules in this region, thereby increasing the relative proportion of crystalline regions within cellulose and improving its relative crystallinity. This change is conducive to enhancing dimensional stability of bamboo [9]. Simultaneously, during heat treatment, the evaporation of adsorbed water in bamboo reduced the distance between cellulose molecular chains in the amorphous region, thereby increasing intermolecular forces and enhancing orientation of cellulose molecules in heat-treated bamboo's amorphous region, ultimately improving dimensional stability [10].

[9] Shi-Cheng Z , Hua-Chun Q I , Yi-Xing L ,et al.Impact on timber crystallization performance of superheated steam-treated wood under high temperature and pressure[J].Journal of Nanjing Forestry University(Natural Sciences Edition), 2010, 34(5):164-166.DOI:10.3724/SP.J.1238.2010.00474.

153

due to less absorption of water

155

You distinguish between the effects in hemicellukose and in cellulose. Are these effcts not the same anyhow?

158

What does this mean: ..water .... disappears during heat treatment?

165

Add standard deviation for these results.

We added the standard deviation in brackets after the test values in the table

167

Add standard deviation for these results.

178

... evaporation of water ...: do you mean that the equilibrium moisture content is lower and therefore the mechanical properties are higher?

The equilibrium moisture content has a great influence on most of the mechanical properties of wood within a certain range. This is also demonstrated in our daily experiments. Sun Yuebo et al. 's research on modified fast-growing poplar shows that when the moisture content of wood changes within the range of 8~15%, the longitudinal compressive strength and bending strength of poplar will increase with the decrease of the equilibrium moisture content. In our study, it was found that the evaporation of adsorbed water in bamboo and the degradation of hemicellulose and cellulose affected the longitudinal compressive strength and bending strength of bamboo together. When the heat treatment pressure was 1.0MPa, the degradation of hemicellulose and cellulose of bamboo was not significant. The change of equilibrium moisture content and density caused by the evaporation of adsorbed water in bamboo becomes the main factor affecting the compressive strength and bending strength along grain. The expression in the paper may not be accurate, so we have revised the expression here:

Based on the FTIR spectra analysis of bamboo samples subjected to heat treatment, it was observed that the degradation level of hemicellulose remained limited under this pressure condition (1.0MPa), which had negligible impact on the mechanical properties of bamboo specimens. The primary change observed under this heat treatment conditions was the evaporation of adsorbed water in bamboo. The evaporation of adsorbed water in bamboo shortens the distance between cellulose molecular chains in the amorphous region, thereby increasing intermolecular forces and promoting tighter arrangement and orderliness. Consequently, this reduces the equilibrium moisture content of bamboo to a certain extent. Below the fiber saturation point, the mechanical properties of bamboo exhibit a negative correlation with its moisture content [11].

189

The reviewer supposes that it is not a synergistic action, but two effect are given, whre the one increases the mechaical properties, whereas the other one decreases them. Up to a certain pressure the "positive" effect is the stronger one, but at higher pressures the degrading effect is stronger.

In this series of heat treatment experiments, the bending strength and longitudinal compressive strength of bamboo were affected by the combined effects of decreasing water content and hemicellulose degradation.

242

"water loss": do you mean reduction of equilibrium moisture content?

Figure 2-3 shows the FTIR spectra of bamboo samples processed by heat treatment under different pressures and bamboo samples from the control group at the wave number of 4000~400 cm-1. The main absorption peaks of the FTIR spectra were assigned as follows: the stretching vibration of the hydroxyl group (—OH) at 3,420 cm-1; the bending vibration of the hydroxyl group (O—H) of adsorbed water in cellulose at 1,630 cm-1; the stretching vibration of the carbonyl group (C=O) of hemicellulose ester at 1,736cm-1; and the stretching vibrations of β-anhydroglucose bonds (C-C-O and C-O-C) at 1,164 cm-1 and 1,035cm-1, respectively. As depicted in Figure 2-3, the peak intensity of the C—H absorption at 1,382 cm-1 remained relatively stable. The bending vibration of the C—H absorption peak was utilized as a reference to determine the relative intensities of other absorption peaks at 3,420 cm-1, 1,736 cm-1, 1,630 cm-1, 1,164 cm-1 and 1,035cm-1, respectively. After heat treatment, the peak intensities of the hydroxyl group in bamboo (3,420 cm-1) and the hydroxyl group of adsorbed water in cellulose (1,630 cm-1) significantly decreased compared to those of the control group as shown in Figure 2-3 and Table 2-3. This suggests that the evaporation of adsorbed water in bamboo occurred during heat treatment. When the pressure increased from 1.0 to 2.5 MPa, there was no significant change in the intensities of absorption peaks at 3,420 cm-1 and 1,630 cm-1, indicating that the variation in pressure had a limited impact on such change processes. It can be deduced that the heat-treatment temperature was still the main factor affecting this change. The peak intensity of the carbonyl group (1,736 cm-1) in the hemicellulose ester of heat-treated bamboo sample showed a decreasing trend compared to that in the control group. This suggests that hemicellulose, which exhibits inferior thermal stability, undergone degradation under hydrothermal conditions resulting in a decrease in its content during heat treatment. Concurrently, the acetyl groups present within the polysaccharide chains of hemicellulose undergo hydrolysis and generate acetic acid[15], which subsequently penetrates into the amorphous region of cellulose and promotes degradation of cellulose molecules within this domain. The decrease in the intensity of characteristic peaks at 1164cm-1 and 1035cm-1, corresponding to C-C-O and C-O-C bonds in β-glucose anhydride, respectively, indicated the occurrence of this process. With an increase in heat treatment pressure, the concentration of carbonyl groups in hemicellulose esters decreased, indicating that increased pressure promoted hemicellulose degradation. Notably, at pressures of 2.0 and 2.5 MPa, hemicellulose degradation was significantly exacerbated during heat treatment. Among the above changes, the evaporation of adsorbed water in bamboo shortens the distance between the cellulose molecular chains in the amorphous region of cellulose, and forms new hydrogen bonds, so that the cellulose molecular chains are more closely arranged and ordered[14], which is conducive to an improvement in bending strength and longitudinal compressive strength of bamboo. And the degradation of hemicellulose and acetyl groups hydrolysis reactions result in a reduction of hydrophilic groups, such as hydroxyl and carbonyl groups, thereby weakening the moisture absorption capacity of bamboo while enhancing its dimensional stability. However, hemicellulose degradation is not conducive to the mechanical properties of bamboo. Among the three main components of bamboo cell wall, hemicellulose acts as a binding agent, wich can fill the gaps between fibers and microfibers [16]. The reduction of hemicellulose weakens the cell wall network skeleton formed by cellulose, thereby reducing the mechanical properties of bamboo. Table 2-3 shows that, When the pressure was set at 1.0 MPa, Only a small amount of hemicellulose was degraded during heat treatment, and the main changes could be ascribed to a evaporation of adsorbed water in bamboo. Compared with the bamboo in the control group, bamboo in the 1.0MPa group showed increased bending strength and longitudinal compressive strength. When the pressures were set at 1.5, 2.0, and 2.5 MPa, the degradation of hemicellulose was gradually aggravated, which caused embrittlement of the bamboo and a decrease in its mechanical properties.

246

This whole section is difficult to read.

As ther reviewer has understood:

a) the pre-treatment reduces the content of hydroxyl groups, in both hemicellulose and cellulose. What is the chemical reaction ongoing? Is it really a formation of ethers (under elimination of water)?

b) Due to the lower content of hydroxyl groups, the hydrophilicity is reduced, hence the equilibrium moisture content is reduced, which increases mechanical strength.

c) With stronger pre-treatment a certain degradation of hemicellulose (and also of cellulose?) takes place, which causes a decrease of the mechanical strengths, despite the fact that the equilibrium moisture content is even lower, which would help to increase strength. But due to the degradation the material becomes softer. Which reactions take place during this degradation of hemicellulose and cellulose? Is there a distinct difference in the degradation reactions between hemicellulose and cellulose?

258

The reviewer supposes that the mechanical properties depend on the crystallinity, not the crystallinity on the mechanical properties.

There is a problem with the wording of this paragraph, and we have amended it as follows:

The crystallinity of bamboo cellulose is closely related to the mechanical properties of the bamboo such as modulus of elasticity and tensile strength, and also affects the hygroscopicity and dimensional stability of bamboo products to some extent.

276

Lines 273-276 are confusing. What is a "quasi-crystalline amourphous zone"?

During the process of heat treatment, hydrolysis occured in the acetyl groups within the polysaccharide molecular chains of hemicellulose, resulting in the production of acetic acid[15]. This acid then entered into the amorphous region of cellulose and promoted degradation of cellulose specifically within this area. As a result, there was an increase in relative proportion of crystalline zones within cellulose [13].

278

What is the standard deviation of the results in Table 2-4? Is the difference in crystallinity statistically significant?

We utilized XRD for testing in the experiment, mainly to verify the degree of influence of heat treatment on the interior of bamboo samples, which was not used as an accurate statistical value, so a large number of samples were not tested by XRD. And we believe that under normal circumstances, because the diffractometer has regular correction, there will be no large sample measurement error in a short time, as long as the sample particle size meets the requirements, we can get more accurate XRD test results.

278

A chapter "Discussion of the results" is missing. There should be much more explanation of what had been measured, about the chemical reactions and reasons behind,

We believe that the increase in relative crystallinity of cellulose is a result of both physical and chemical changes occurring during heat treatment. On the one hand, The evaporation of adsorbed water in the amorphous region of cellulose reduces the distance between cellulose molecular chains, leading to the formation of new hydrogen bonds. This results in an increase in intermolecular forces and a more tightly arranged and ordered structure, ultimately resulting in an increase in relative crystallinity of bamboo cellulose. On the other hand, The acetic acid produced by the hydrolysis of acetyl groups in hemicellulose polysaccharide chains infiltrates into the amorphous region of cellulose, thereby promoting degradation in this area and increasing the relative proportion of crystalline cellulose zones. As evidenced by the data presented in Table 2-4, the heat treatment applied under this experimental conditions primarily targets the amorphous region of bamboo cellulose, while leaving the crystalline region unaffected. This finding is consistent with previous research on high temperature heat treatments of bamboo. From an economic standpoint, this study highlights the potential value of further exploring medium and low temperature heat treatment processes.

287

Sentence 286-289 needs reformulation; it tell twice the same.

In this study, the heat-treatment time and temperature were kept constant while systematically investigating the effects of different pressures on bamboo's equilibrium moisture content, dimensional stability, mechanical properties, and chemical composition. The experimental results demonstrated that: (1) According to the FTIR and XRD spectra of bamboo samples, physical and chemical changes occurred during heat treatment in this experiment. These changes included evaporation of adsorbed water in bamboo, degradation of hemicellulose and cellulose, as well as hydrolysis of acetyl groups in the molecular chains of hemicellulose. Among these changes, the evaporation of adsorbed water was mainly affected by the temperature used for heat treatment. The variation in heat treatment pressure has negligible impact on it. However, the extent of hemicellulose and cellulose degradation as well as acetyl groups hydrolysis within hemicellulose molecular chains were enhanced with increasing heat treatment pressure. (2) With the increase of heat treatment pressure, there is a gradual decline in the equilibrium moisture content of bamboo. The physical and chemical changes in this experiment weaken the hygroscopic properties of bamboo, with hemicellulose degradation and acetyl group hydrolysis being the main influencing factors. (3) With the increase of heat treatment pressure, the dimensional stability of bamboo was improved, that is also due to the occurrence of the above physical and chemical changes, which weakens the moisture absorption ability of bamboo and increases the relative crystallinity. (4) With the increase in heat treatment pressure, the bending strength and longitudinal compressive strength of bamboo initially increased before subsequently decreasing. This change was jointly affected by the evaporation of adsorbed water in bamboo and the degradation of hemicellulose and cellulose. These physical and chemical changes reduce the equilibrium water content while increasing the relative crystallinity of bamboo, which is conducive to an increase in its mechanical strength. However, the degradation of hemicellulose and cellulose simultaneously resulted in the destruction of the bamboo cell wall's skeletal structure, rendering it fragile and ultimately reducing its mechanical strength. When subjected to a heat treatment pressure of 1.0 MPa, only a minimal amount of hemicellulose and cellulose undergone degradation during the process, instead, evaporation of adsorbed water within the bamboo is primarily responsible for any changes observed at this stage - which ultimately serves to enhance its mechanical strength. At this stage, the bamboo exhibits its highest bending strength and longitudinal compressive strength, measuring 41.2MPa and 42.6MPa respectively. As heat treatment pressure increases further, hemicellulose and cellulose degradation intensifies, leading to greater damage of the bamboo cell wall and a gradual decrease in both bending strength and longitudinal compressive strength.

292

But the real reason is that the equilibrium moisture content is lower, and therefore the mechanical strengths are higher, at least at still moderate pre-treatment.

293

Explain the reasons for this fact.

Round 2

Reviewer 1 Report

The paper has much improved with this first revision.Nevertheless there are still several details open, which need to be clarified and explained, see attached reviewed version of the paper.

Some editing of language is necessary.

Author Response

Dear Editor

     Thanks for you and the reviewers' positive work about our manuscript. We have revised our manuscript according to the comments. I think the quality of this manuscript improved greatly.

Best regards!

 Jialei Liu

Reviewer 2 Report

The manuscript has been sufficiently improved to warrant publication in Polymers.

Author Response

Dear reviewer

         Thanks for your positive work about our manuscript. The quality of our manuscript was improved greatly under your help.

Best regards!

 Jialei Liu

Round 3

Reviewer 1 Report

Paper has improved, but still some questions rename open and need clear description and explanation. See attached reviewed paper.

See above.

Author Response

Dear reviewer, thank you so much for your valuable suggestions and insightful comments on our paper again. We have carefully addressed all of your inquiries and made the necessary revisions to the manuscript based on your feedback. The changes are shown in green in the article.
